# A Novel Multi-Cell Interference-Aware Cooperative QoS-Based NOMA Group D2D System

**Muhammad Amish Hasan** [1,†], **Tanveer Ahmad** [2,*,†] , **Asim Anwar** [1,†], **Salman Siddiq** [1,†], **Abdul Malik** [1,†], **Waseem Nazar** [1,†] **and Imran Razzaq** [1,†]

1   Department of Technology, The University of Lahore, Lahore 54590, Pakistan
2   Innovation Education and Research Center for On-Device AI Software (Bk21), Department of Computer Science and Engineering, Chungnam National University, Daejeon 34134, Republic of Korea
*   Correspondence: tahmad@cnu.ac.kr
†   These authors contributed equally to this work.

**Abstract:** Nonorthogonal multiple access (NOMA), one of the favorable candidates of next-generation wireless networks combined with group device-to-device (D2D) networks, can sufficiently increase a system's spectral efficiency. In fact, in a cooperative scenario, successive interference cancellation (SIC) is used in NOMA receivers to reduce the complexity of relaying, as each user has to decode high-order user data. This work presents a quality of service (QoS)-based cooperative NOMA-aided group D2D system (Q-CNOMA). The Q-CNOMA system not only reduces the burden on the group transmitter by relaying the signal to a receiver in neighboring cells but also improves the overall system performance. In order to model the major components in a D2D scenario such as receivers clustering around a transmitter, the spatial distribution of D2D transmitters is modeled using a Gaussian–Poisson process (GPP). A closed-form expression of outage probability is calculated and benchmarked against conventional systems to prove the superiority of the proposed Q-CNOMA system.

**Keywords:** quality of service; device-to-device communication; non-orthogonal multiple access; stochastic geometry

## 1. Introduction

The next generation (5G and beyond, 6G) of wireless networks promises to deliver high reliability, increased data rate, and low latency under diverse and ubiquitous connectivity scenarios. The current applications of mobile communication have risen beyond simple voice and data communication to data-hungry, delay-sensitive applications. As such, researchers from academia and industry have defined many key enabling technologies in order to meet the stringent quality-of-service (QoS) requirements of next-generation wireless networks [1]. However, the role of multiple-access schemes is always pivotal in enhancing the overall system performance. As a consequence, non-orthogonal multiple access (NOMA) is proposed as a promising multiple-access scheme in order to address the challenges posed by 5G, and beyond, wireless networks [2].

NOMA can be modelled in pattern division multiple access [3], sparse-code domain [4], power domain [3] and low-density spreading [5]. A different power level is assigned to each user on the basis of channel conditions for data transmission in the NOMA power-domain technique. As such, multiple users are superimposed onto the same resource (time/frequency/code/power, etc.), resulting in an overall improvement in spectral efficiency.

Another attractive feature of NOMA is that it can be easily integrated into other enabling technologies proposed for next-generation wireless networks, such as device-to-device (D2D) [6], multiple input multiple output (MIMO) [7], millimetre wave [8], sensor networks [9] and full-duplex (FD) communication [10]. In fact, D2D communication is another technique that can be used to enhance the spectral efficiency of radio systems by enabling cellular users to communicate independently with neighboring users in a D2D

communication network [11]. Furthermore, cooperative relaying combined with NOMA possesses a significant potential to enhance the network throughput. In particular, by employing cooperative relaying NOMA, the data rate of the far users would be expected to improve without the involvement of a base station [12]. Hence, introducing D2D communication along with NOMA relaying is of paramount importance.

### 1.1. Related Work

Recently, a lot of research works have been performed on NOMA and D2D relaying separately. However, little attention has been paid to relaying in NOMA-based D2D communication. The authors in [13] proposed a two-phase cellular and D2D transmission scheme which is based on the principle of NOMA. The authors considered a single-cell network with two users (near and far), where the base station (BS) transmits data simultaneously to both the users. However, based on the decoding status, either the D2D transmitter or receiver send the BS data to the far user. In order to demonstrate the effectiveness of the proposed scheme, the authors conducted extensive simulations to analyze the overall throughput of the system. The results show that the proposed system outperforms the existing systems, which are based on the OMA technique. However, the system only considers two users and is limited to only a single-cell scenario. In [14], the authors proposed a joint subchannel and power allocation algorithm for a NOMA-associated D2D communication system. The authors proposed a novel many-to-one matching-theory-based subchannel allocation algorithm which was then used to solve the non-convex power-allocation problem. The simulation results show that the sum rate of the NOMA-associated D2D systems is better than the OMA-based D2D system. However, the authors only considered a single cell without corporation between the users.

In [15], the authors considered an OMA-based full-duplex (FD) cooperative D2D (C-D2D) network. The D2D transmitter (DT) assumes the role of FD relay in order to realize bidirectional communication between the BS and cellular user. The authors employed NOMA in order to manage resources between cellular and D2D users. Further, the overall system performance was evaluated by deriving the analytical expressions for achievable rates (cellular and D2D users) and outage probability (FD and half-duplex (HD) systems). The numerical results demonstrate that the proposed system manages to achieve better outage performance for both the cellular and D2D user. However, the proposed system lacks analysis of its interference and is also limited to the single-cell scenario. The authors in [16] explored a NOMA-based cooperative cellular network. In the proposed network setting, only the center cell users can establish a direct communication link with the BS, whereas the edge cell users would require a relay in HD mode to facilitate its transmission. Further, a new two-stage relay selection strategy (TSRS) was proposed which aims to reduce the outage probability of center cell users while enhancing the probability of successful reception at the edge cell user. The authors derived the closed-form analytical expressions for outage probability at each user (center and edge) in order to evaluate the performance. The results show that the proposed D2D-aided cooperative NOMA system achieves better outage performance than the conventional partial-relay selection scheme.

The authors in [17] introduced the idea of NOMA-based group D2D communication. The system uses a power allocation scheme to reduce the interference between different D2D groups using the same sub-channels. The authors only considered a single-cell scenario which lacks the effect of interference, and the spatial distribution of the D2D does not follow any specific distribution. A group D2D-enabled system model in a NOMA communication system was proposed in [18], and the authors calculated the outage performance of the system in a multi-cell scenario, but the cooperation between the nodes, which would further improve the system's performance, was not analyzed. In [19], the authors proposed a cooperative full-duplex (FD) D2D-based NOMA communication network. The strong and weak users in a single cell co-operate with each other to increase the outage performance of the system. The base station sends the signal to an FD-enabled strong user and forward it to its predefined weak user pair. The authors in [19] only consider a single-cell scenario,

i.e., no interference characterization at the receiver and no spatial topology was considered for D2D users. The effect of the integration of D2D in a downlink NOMA system was investigated in [20]. The authors proposed power control techniques for a pair of D2D users underlaid with a pair of cellular users. The superiority of NOMA over OMA was shown in terms of sum rate. An uplink multi-carrier NOMA-based D2D system underlaid with a cellular network was considered in [21]. The authors proposed an iterative algorithm using the Karush–Kuhn–Tucker (KKT) condition to enhance the channel throughput of the system. The system achieves near optimal performance. However, device clustering of the D2D network was ignored. In [22], a D2D-aided cooperative relaying system employing NOMA to increase the spectral efficiency of the system was proposed. In the proposed model, the base station transmits a signal to the relay and receivers. To increase the spectral efficiency of the system, the receiver obtains another independent signal from the relay. The relay also transmits its signal to another receiver, thus further enhancing the spectral efficiency. A power allocation strategy was also proposed, to enhance the capacity and the signal-to-noise ratio (SNR). However, when considering a multi-cell scenario, their results cannot be generalized and no particular random distribution was followed to define the spatial distribution of D2D users. The authors in [23] developed a sector-based green NOMA D2D system. A novel multiple-interference cancellation scheme was proposed which minimizes the interference in the network with optimized resource allocation. The proposed system is inferior to the SIC-based NOMA system. However, some performance gains in terms of energy efficiency and fairness factor were achieved.

A D2D-assisted NOMA relaying scheme was proposed by the authors of [24]. The authors improved the signal-to-interference-noise ratio ($SINR$) of a far user by employing maximal ratio combining (MRC) on a NOMA signal in the first phase of communication from a base station and in a D2D-assisted NOMA signal from relay in the second phase. The proposed system achieves a better average data rate than the traditional OMA system. However, the user location is assumed to be fixed and no interference is assumed at the receivers, which make the proposed system very simple and unrealistic. The authors in [25] investigated an under-laid NOMA-based D2D network. The sum rate of the overall system is maximized by employing a joint power allocation and user clustering. The proposed model achieves performance gain in terms of average sum rate and user connectivity as compared to a traditional NOMA and OMA system. The proposed model deploys the D2D users at fixed locations and follows PPP, which cannot fully comprehend a D2D network [18,26]. A cooperative NOMA-associated D2D system employing decode and forward relaying was considered in [27]. The authors assumed a direct link between the base station and D2D user and an extra power-line link for weak users. The system was analyzed for the optimum and operational value of the power allocation coefficient for the strong user. The simulation results showed that the outage probability reduced sufficiently with the addition of a power-line link at low SNR values. However, a multi-cell corporation analysis was not included in the discussion. In [28], the proposed system's model employs the NOMA power domain to separate the user data in a large-scale D2D network using a cooperative hybrid automatic repeat request (HARQ). The system performance in terms of outage and throughput was studied and it was shown that the proposed system outperforms OMA and non-cooperative NOMA. A potential limitation in this system's model is that the authors only consider two user NOMA transmissions in D2D communication. The power allocation in the NOMA signal was based on the assumption that only a single user is close to the source, which is not true as D2D receivers always tend to cluster around their respective transmitter. This aspect makes the channel condition of the users very similar to each other. Furthermore, the proposed D2D network is modelled by a poisson point process (PPP), which is not sufficient to fully model the behaviour of a D2D network.

*1.2. Motivation and Contributions*

The research towards the performance analysis of cooperative relaying in NOMA D2D-based systems in multi-cell scenarios is still under consideration; especially, the behaviour

of D2D devices is not often considered. In the literature [18,26,29], PPP is commonly used to analyze the behaviour of D2D networks. However, PPP is inferior to the Gaussian–Poisson process (GPP) when clustering the behaviour of D2D networks is under consideration. GPP is a class of cluster point processes which have a simple structure as compared to PPP and also provide a better modelling in a cooperative environment [29]. In this paper, a quality-of-service-based cooperative NOMA D2D network (Q-CNOMA) in a multi-cell interference-aware environment is proposed. The transmitters in the proposed system model, i.e., the D2D group transmitters, are distributed randomly over $\mathbb{R}^2$ following GPP, and the D2D receivers are randomly clustered around the transmitter.

Another important aspect of NOMA D2D communication networks is power allocation among users. The NOMA in a D2D system assigns the power to users depending on their channel conditions. This ordering of the user is not optimal for group D2D users as the D2D receivers are almost always in close proximity, which leads to the same channel conditions resulting in similar power allocation coefficients. This approach limits the benefits of the NOMA in D2D communication. In contrast to ordering on the basis of channel condition, few authors [16,30] considered ordering the user according to their QoS. The proposed Q-CNOMA system uses user's desired rate as the QoS measure to assign the power allocation coefficients.

The limitations and gaps in the aforementioned discussion prompted us to evaluate and explore the performance of cooperative relaying in NOMA-aided group D2D networks. In this work, we propose and investigate group D2D communication with a Q-CNOMA network and try to overcome the above-mentioned limitations and gaps. To the best of our knowledge, this is the first time that Q-NOMA has been analyzed in a cooperative group D2D relaying in a multi-cell scenario.

The primary contributions of this work can be summarized as follows.

- We propose a multi-cell cooperative relaying in Q-NOMA-enabled D2D communications. To completely analyze the device clustering and spatial separation, GPP is used to model the D2D transmitter topology, and the D2D receivers are randomly distributed around it.
- The power allocation coefficients are assigned depending upon the QoS of users.
- Furthermore, closed-form expressions of interference and the outage probability at the D2D receiver are derived. The interference approximation for cooperative Q-NOMA-aided D2D relaying is used to derive the closed-form expression of outage probability.
- Analytical and simulation results are established to validate the outage probability results and to demonstrate the dominance of Q-NOMA over conventional OMA-assisted D2D systems.

### 1.3. Gauss–Poisson Process

The GPP can be defined as a homogeneous Poisson cluster process in which clusters exhibit independence. Based on the parent-process intensity, denoted as $\lambda$, the clusters in GPP can be characterized as single- or two-point clusters with probabilities $1 - a$ and $a$, respectively. In case of a single-point GPP, the cluster is located at the position of the parent. However, when GPP is composed of two points, one is designated as a parent point and the second point is randomly distributed around the parent point [29].

The proposed system model is described in Section 2. The closed-form expression of the outage probability at the final receiver under a multi-cell interference is presented in Section 3. The performance of the proposed system and the impact of different parameters on outage probability is presented in Section 4. Finally, the paper is concluded in Section 5.

## 2. System Model

Consider multi-cell cooperative relaying in Q-NOMA-enabled D2D scenario shown in Figure 1. It is assumed that each group transmitter sends data to its clustered devices using NOMA and relay nodes ($R_e$s) are supposed to send data to the adjacent cells' group transmitters using OMA. In our case, the device connected to group transmitter one ($G_{T_1}$)

wants to transmit data to a device present in neighboring cell connected to its group transmitter $(G_{T2})$. As such, under this particular setting, the entire communication is divided into two phases. $G_{T_1}$ in Phase I is responsible for transmitting the NOMA signal to all the receivers $(r_x)$ in its coverage area (any receiver would act as a $R_e$ in Phase II). In Phase II, the relay nodes communicate to $(G_{T2})$ of the concerned receiver using OMA transmission. The signal received at the $i$th relay in Phase I is written as:

$$y_{R_e,i} = h_{i,1} \sum_{m_1=1}^{M_1+1} \sqrt{\beta_{m_1} P_{G_{T_1}}} S_{m_1} + n_{i,1} + I$$

$$i = 1, 2, \cdots, M_1$$

(1)

where $h_{i,1} = \frac{\tilde{h}_{i,1}}{\sqrt{1+d_{i,1}^{\alpha}}}$ denotes the channel between $i$th relay and $G_{T_1}$ during Phase I, $\tilde{h}_{i,1}$ is the Rayleigh fading channel gain, $d_{i,1}$ is the distance of the $i$th relay to the $G_{T_1}$, $\alpha$ is the path loss factor, $\beta_{m_1}$ is the power allocation coefficient, and $S_{m_1}$ is the message signal of $m_1$th receiver. $P_{G_{T_1}}$ denotes the transmission power of $G_{T_1}$, $I$ is the interference, and $n_{i,1}$ is the additive white Gaussian noise (AWGN) with zero mean and variance $\sigma^2$. Any number of $r_x$ near the cell edge can act as a D2D $R_e$ and forward the desired data to $G_{T_2}$ (group transmitter in Phase II). The relays receive $M_1 + 1$ data streams multiplexed using NOMA and decode two data signals, one data stream being its own signal and the second stream the data that needs to be relayed $(S_{R_e})$. In Phase II, each $R_e$ will transmit the decoded data to $G_{T_2}$. The $G_{T_2}$ will employ MRC and transmits the data to $R_x$ using NOMA as follows:

$$\begin{bmatrix} y_{2,1} \\ \vdots \\ y_{2,i} \end{bmatrix}^T = \begin{bmatrix} h_{2,1} \\ \vdots \\ h_{2,i} \end{bmatrix}^T S_{R_e} P_{R_e} + \begin{bmatrix} n_{2,1} \\ \vdots \\ n_{2,i} \end{bmatrix}^T$$

$$\hat{y}_{G_{T_2}} = \mathbb{H} S_{R_e} P_{R_e} + \tilde{n}$$

(2)

where $i = \{1, 2, \cdots, M_1\}$, $y_{2,i}$ represents the received signal at a $G_{T_2}$ transmitted from $i$th relay, $h_{2,i}$ and $n_{2,i}$ are the Rayleigh fading channel gain and AWGN between $i$th relay and $G_{T_2}$, respectively. For simplicity, we assumed that every relay transmits signal with power $P_{R_e}$. After MRC, the received signal at $G_{T_2}$ can be written as:

$$y_{G_{T_2}} = \frac{\mathbb{H}^H S_{R_e} P_{R_e} + \mathbb{H}^H \tilde{n}}{||\mathbb{H}||}$$

$$= ||\mathbb{H}|| S_{R_e} P_{R_e} + \frac{\mathbb{H}^H \tilde{n}}{||\mathbb{H}||}$$

(3)

$G_{T_2}$ is responsible for delivering the message signal to all $r_x$ in its cell. The received signal at $R_x$ can be denoted by $y_{R_x}$ and is written as:

$$y_{R_x} = h_{R_x,2} \sum_{m_2=1}^{M_2} \sqrt{\beta_{m_2} P_{G_{T_2}}} S_{m_2} + n_{R_x,2}$$

(4)

After SIC, the desired signal at $R_x$ can be written as follows:

$$y_{R_x} = h_{R_x,2} \left( \sqrt{\beta_{R_x} P_{G_{T_2}}} ||\mathbb{H}|| S_{R_e} P_{R_e} + \frac{\mathbb{H}^H \tilde{n}}{||\mathbb{H}||} \right) + n_{R_x,2}$$

$$= h_{R_x,2} \left( \sqrt{\beta_{R_x} P_{G_{T_2}}} ||\mathbb{H}|| S_{R_e} P_{R_e} \right) + \aleph^2$$

(5)

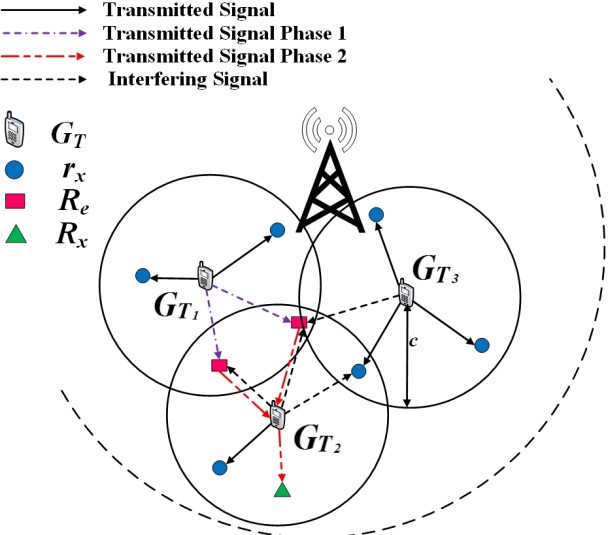

**Figure 1.** Proposed system model.

Here, $\aleph = \sqrt{h_{R_x,2}\sqrt{\beta_{R_x}P_{G_{T_2}}}\left[\frac{\mathbb{H}^H\tilde{n}}{||\mathbb{H}||}\right] + n_{R_x,2}}$. Rayleigh fading channel gain and AWGN between $R_x$ and $G_{T_2}$ are denoted by $h_{R_x,2}$ and $n_{R_x,2}$, respectively. In group D2D network, $r_x$ are located in close proximity to each other and exhibit clustering around their corresponding $G_T$. The channel conditions of these $r_x$ are very similar to each other; hence, when employing NOMA, the ordering of users on the basis of their channel gains is not optimal [18]. To achieve the desired multiplexing gains and the fairness among receivers, we ordered the users on the basis of their QoS requirement which is the desired rate in our case. Power allocation schemes used in our work are presented in the next section.

*Power Allocation Scheme*

In NOMA, different users are separated on the basis of their respective power, making the power allocation coefficient a major factor in affecting the performance of the system. In proposed Q-CNOMA-based system, the ordering of users is based on their respective user rate $\mathbb{B}_i$ not on their channel conditions, as users are placed very close to each other. The power allocation scheme employed in our system uses rates of receivers to assign the power allocation coefficient which can be calculated as follows according to [18]:

$$\beta_i = \frac{1/\mathbb{B}_i}{\sum_{j=1}^{U}\frac{1}{\mathbb{B}_j}} \tag{6}$$

where $U$ is the total number of users in a cell and $\mathbb{B}_i$ is the rate of the $i$th user. In order to maintain the fairness among users, inverse rates are used for power allocation so that the user with a high target rate does not experience a high $SINR$ as compared to users with low rate requirement.

## 3. Outage Probability Analysis

This section presents the analysis of outage probability for the $R_x$ in proposed Q-CNOMA-based D2D relaying system. Note that from (5), the received SINR/signal depends on interference. As such, in order to compute the outage probability, the interference distribution is required. Therefore, before deriving the outage probability, the interference distribution is calculated using the Lemma 1, as given below.

Now with interference distribution in hand, we can proceed towards finding the outage probability. In order to write the outage event at $R_x$, let us first define the following two events:

$$\mathcal{A}_1 = \{\text{Outage in Phase I}\} \tag{7}$$

$$\mathcal{A}_2 = \{\text{Success in Phase I}\}\text{and}\{\text{Outage in Phase II}\} \tag{8}$$

Note that the events $\mathcal{A}_1$ and $\mathcal{A}_2$ are mutually exclusive. As such, based on (7) and (8), the overall outage probability at the $R_x$ in the considered network, denoted as $\mathbb{P}_{out}$, can be written as follows:

$$\mathbb{P}_{out} = \mathbb{P}_1 + (1 - \mathbb{P}_1)\mathbb{P}_2 \tag{9}$$

where $\mathbb{P}_1$ and $\mathbb{P}_2$ represent the outage probability of Phase I and Phase II, respectively. Subsequently, the outage probabilities $\mathbb{P}_1$ and $\mathbb{P}_2$ are expressed in following Lemma and Theorem, respectively.

**Lemma 1.** *The outage probability at the receiver in Phase I can be obtained using the results presented in [2] and can be expressed as follows:*

$$\mathbb{P}_1 = \sum_{l=1}^{L} b_l e^{\phi_{m,1}^{max} \psi_l} \mathcal{L}_I \left( \frac{\phi_{m,1}^{max} \psi_l \rho}{\rho_t} \right) \tag{10}$$

*where $b_l = \omega \sqrt{1 - \theta_l^2}(1 + \theta_l)$, $\omega = \frac{\pi}{2}$, $\psi_l = \left(\frac{c}{2}(1 + \theta_l)\right)^\alpha$, $\theta_l = \cos\left(\frac{(2l+1)\pi}{2L}\right)$, c represents the radius of coverage area, $\mathcal{L}_I$ denotes the Laplace transform of the interference I, L is the complexity–accuracy trade-off parameter, $\rho = \frac{P_I}{\sigma^2}$, $P_I$ is the maximum transmission power available to interfering transmitters, $\rho_t = \frac{P_{G_{T_1}}}{\sigma^2}$, $\phi_{m,1}^{max} = \max(\phi_{1,1}, \phi_{2,1}, \cdots, \phi_{m,1})$, $\phi_{j,1} = \frac{\tau_{j,1}}{\beta_i - \tau_{j,1} \sum_{m,1=j,1}^{M_1+1} \beta_{m,1}}$, $\tau_{j,1} = 2^{\mathbb{B}_{j,1}^{th}} - 1$, and $\mathbb{B}_{j,1}$th is the targeted rate of jth user in Phase I.*

**Theorem 1.** *The outage probability at the receiver in Phase II can be expressed as follows:*

$$\begin{aligned}
\mathbb{P}_2 =& 1 - \exp^{-\gamma_t/\Omega_{2,i}} + \sum_{l=1}^{L} \sum_{k_1=0}^{N_1} \sum_{k_2=0}^{2N_2} \sum_{k_3=1}^{N_3} \frac{b_l}{T} e^{-\eta_1} \\
& \cdot \frac{(-1)^{k_1}}{k_1!} (\psi_l \eta_2)^{k_1} \Gamma(1 - k_1, \tilde{\gamma}_t) \omega_{k_3} e^{x_{k_3}} \\
& \cdot (1 + x_{k_3}\xi_I)^{k_1} \Re\left[ \mathcal{L}_I(s) e^{(\gamma + \frac{l\pi}{T})x_{k_3}} \right]
\end{aligned} \tag{11}$$

*where $\gamma_t = \frac{\tau_{R_x}}{\xi_{R_x}}$, $\Omega_{2,i}$ is the variance of the Rayleigh CDF, $\tau_{R_x} = 2^{\mathbb{B}_{th}^{R_x}} - 1$, and $\xi_{R_x} = \frac{P_{G_{T_2}}}{\aleph^2}$; L, $N_1$, $N_2$, and $N_3$ are the complexity–accuracy trade-off parameters; $\xi_I = \frac{1}{\aleph^2}$; $\eta_1 = \frac{\tau_{R_x}}{\beta_{R_x}} \sum_{m_2=j_2}^{M_2} \beta_{m_2}$; $\eta_2 = \frac{\tau_{R_x}}{\xi_{R_x}\beta_{R_x}}$; $\beta_{m_2}$ is the power allocation coefficient of user $m_2$; $M_2$ are the total number of users in Phase II, and $\Gamma(\cdot)$ is the Gamma function.*

Proof: Please see Appendix A.

## 4. Numerical Results

This section presents the numerical results in order to validate the accuracy of the analytical outage probability expression derived in Equations (9)–(11) of Section 3 under the proposed system model presented in Figure 1. In particular, the Monte-Carlo method was utilized to obtain simulation results. In addition, Matlab software is used to generate the numerical results as well as to carry out the Monte-Carlo simulations. Further, in all the results, unless otherwise stated, the default power allocation scheme presented in

Equation (6) was adopted. In all simulations, the parameters presented in Table 1 were considered, unless otherwise specified.

**Table 1.** Simulation parameters.

| Parameter | Description | Value |
|---|---|---|
| $M_1$ | Number of users in Phase I | 2 |
| $M_2$ | Number of users in Phase II | 2 |
| $N_{R_e}$ | Number of relays | 2 |
| $\{\mathbb{B}_j\}_{j=1}^{M_1}$ | Users' rates in Phase I | {0.7, 1.2} |
| $\{\mathbb{B}_j\}_{j=1}^{M_2}$ | Users' rates in Phase II | {0.5, 1.1} |
| $c$ | Radius of coverage area | 10 m |
| $\alpha$ | Path loss exponent | 4 |
| $\lambda_{GTx}$ | Intensity of group transmitters | $10^{-4}$ |
| $d_1$ | Distance of group transmitter from relay in Phase I | 9 m |
| $d_2$ | Distance of group transmitter from relay in Phase II | 8 m |
| $P$ | Degree of Gauss–Laguerre polynomial | 5 |
| L, N, V, Q, S | Gaussian–Chebyshev parameters | 5 |

### 4.1. Outage Probability Comparison

In Figure 2, the average outage probability of the proposed system is benchmarked against the relevant systems found in the literature, including the QoS-based group D2D NOMA communication network. In Figure 2, the proposed Q-CNOMA system shows a remarkable performance gain over all SINR regimes. However, the distance between the receiver and transmitter increases, and that results in an increase in the system's outage probability. In the paired D2D OMA system, the transmission requires many relay nodes, not only increasing the total time slots but also affecting the OMA transmission's effect on the performance of the system. The burden of calculating the power allocation coefficient at the group transmitter is ignored when using the fixed power allocation policy with the proposed system. However, it affects the outage of the relay networks, thus degrading the systems' performance.

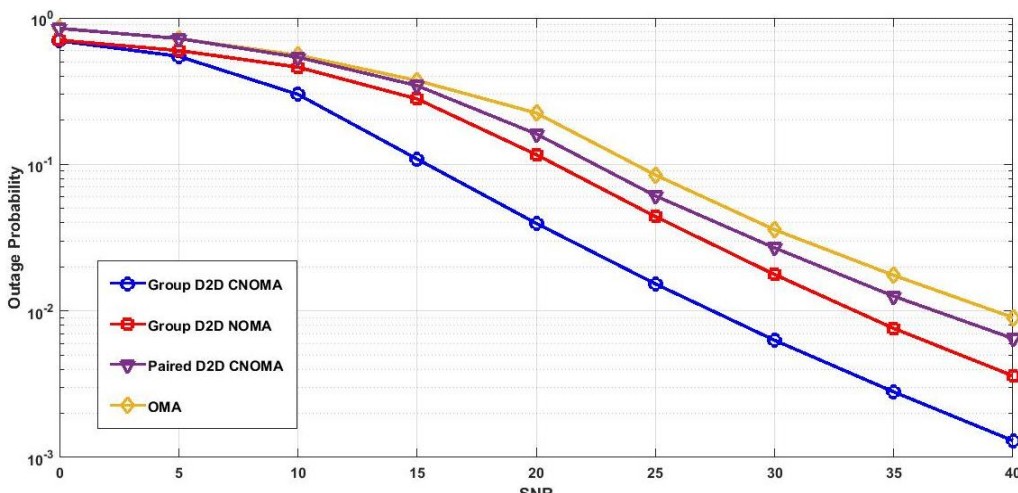

**Figure 2.** Outage probability comparison of proposed Q-CNOMA with C-NOMA, Q-NOMA, NOMA and OMA.

### 4.2. Impact of $M_1$ and $R_e$ on Outage Probability

The impact of varying the number of users in Phase I while varying the number of relays is investigated in Figure 3. The Monte-Carlo simulation was carried out to verify the analytical expression of the outage probability. The simulation and analytical results are in good agreement, validating the performance analysis. Results clearly show that increasing

the number of users in a cell with NOMA transmission would increase the average outage probability of the system. However, it can be observed that increasing the number of relay nodes would improve the performance of the system.

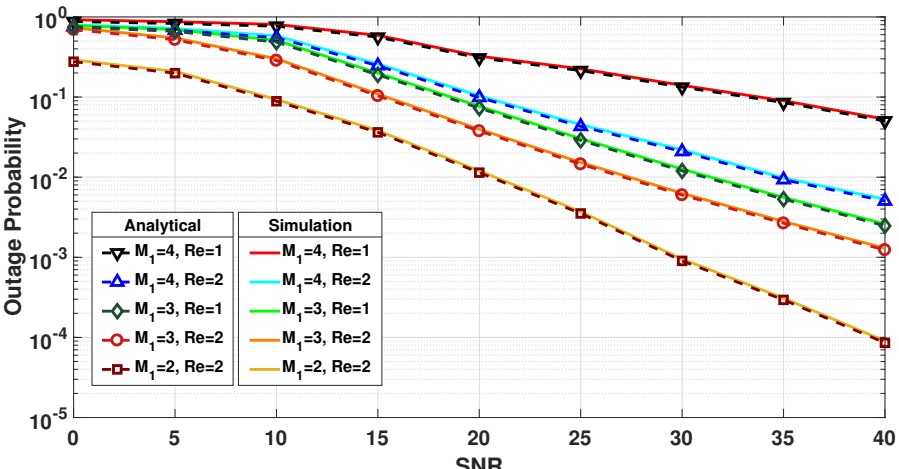

**Figure 3.** Impact of $M_1$ and $R_e$ on outage probability of proposed Q-CNOMA system.

### 4.3. Impact of $M_2$ and $\alpha$ on Outage Probability

Figure 4 describes the average outage probability of the proposed model with varying the number of users in Phase II for different path loss values. It can be seen that the system achieves lower average outage probability as the number of users in Phase II deceases. This phenomenon is experienced because the allocation of the power coefficients in the NOMA system is based on the user's channel condition or desired rate. Therefore, increasing the number of users results in power allocation coefficients that are relatively close to each other making it difficult for SIC to perfectly decode each user's data.

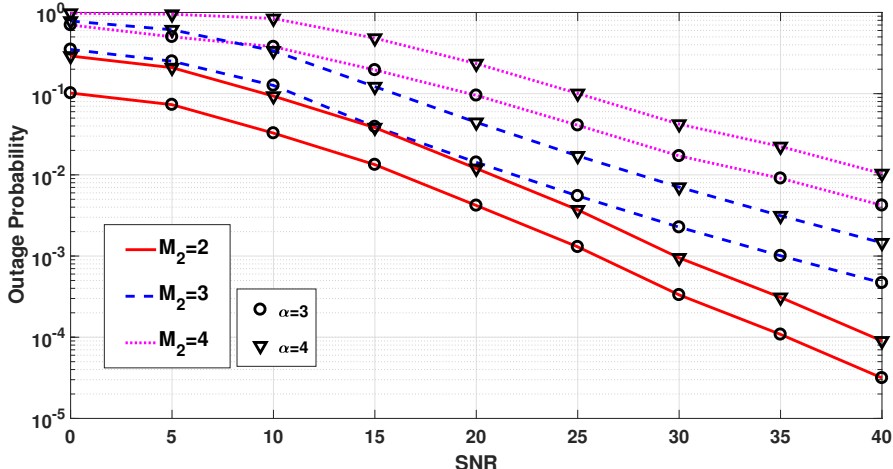

**Figure 4.** Impact of $M_2$ and $\alpha$ on the outage probability of the proposed Q-CNOMA system.

## 5. Conclusions

In this paper, we investigated a QoS-based D2D NOMA system in a multi-cell interference-aware cooperative environment. The topology of D2D transmitters was modeled by GPP. As the users are located in the close proximity to each other, the power allocation coefficients were modeled on user data rate instead of channel conditions. A closed-form expression for the outage probability of Q-CNOMA was derived for the end receiver when the power allocation is based on users' desired rates. Simulation results further verify the correctness of the analytical results with Monte-Carlo simulations. Sim-

ulation results demonstrate that the proposed Q-CNOMA system achieves better outage probability than a Q-CNOMA system with fixed power allocation coefficients, Q-NOMA system with enlarged coverage area, NOMA-based system, and paired D2D OMA system. The simulation results further elaborate that increasing the number of users in Phase I increases the outage. However, this effect is reduced by introducing relay nodes in the same cell. As compared to the number of users in Phase I, the outage probability is more affected by increasing the users in Phase II, as verified by simulations. The proposed scheme can have prominent applications in future sensor networks where nodes have a limited power source. Future works can be carried out on relay selection in the proposed Q-CNOMA system, to further enhance the system's performance. Another interesting dimension is to investigate the current system underlying the cellular system with multiple antenna transceivers.

**Author Contributions:** Conceptualization, M.A.H. and A.A.; Methodology, M.A.H. and A.A.; Software, T.A., Formal analysis, A.A., T.A., S.S. and A.M.; Investigation, T.A., A.A., W.N. and M.A.H.; Resources, A.A., A.M. and I.R.; Data curation, T.A.; Project administration, T.A., S.S. and A.A.; Funding acquisition, T.A. All authors have read and agreed to the published version of the manuscript.

**Funding:** This research received no external funding.

**Data Availability Statement:** Data is available upon request from the corresponding author.

**Conflicts of Interest:** The authors declare no conflict of interest.

**Appendix A. Outage Probability in Phase II**

**Proof.** In Phase II, the relays transmit their data to $G_{T_2}$, which is responsible for delivering it to the end receiver, i.e., $R_x$. The $SINR$ of the received signal given in (5) can be written as:

$$SINR_{R_x} = \frac{h_{R_x,2}||\mathbb{H}||^2 \beta_{R_x} P_{G_{T_2}}}{h_{R_x,2} \sum_{m_2=j_2}^{M_2} \beta_{m_2} P_{G_{T_2}} + I + \aleph^2} \tag{A1}$$

$$j_2 = 1, 2, \cdots, M_2$$

The users in the proposed system's model are in close proximity to each other. The outage event at $R_x$ $(\varphi_{R_x})$ occurs if the threshold rate of any high-order user or the threshold rate of $R_x$, $\mathbb{B}_{th}^{R_x}$ is not achieved and can be described as follows:

$$\begin{aligned}
\varphi_{i \to R_x} &= \left\{ \mathbb{B}_{i \to R_x} < \mathbb{B}_{th}^{R_x} \right\} \\
&= \left\{ \log_2 \left( 1 + \frac{h_{R_x,2}||\mathbb{H}||^2 \beta_{R_x} P_{G_{T_2}}}{h_{R_x,2} \sum_{m_2=j_2}^{M_2} \beta_{m_2} P_{G_{T_2}} + I + \aleph^2} \right) < \mathbb{B}_{th}^{R_x} \right\} \\
&= \left\{ \frac{h_{R_x,2}||\mathbb{H}||^2 \beta_{R_x} P_{G_{T_2}}}{h_{R_x,2} \sum_{m_2=j_2}^{M_2} \beta_{m_2} P_{G_{T_2}} + I + \aleph^2} < \tau_{R_x} \right\} \\
&= \left\{ h_{R_x,2} < \frac{\tau_{R_x}(\xi_I I + 1)}{\xi_{R_x}(||\mathbb{H}||^2 \beta_{R_x} - \tau_{R_x} \sum_{m_2=j_2}^{M_2} \beta_{m_2})} \right\}
\end{aligned} \tag{A2}$$

where $\xi_I = \frac{1}{\aleph^2}$, $\tau_{R_x} = 2^{\mathbb{B}_{th}^{R_x}} - 1$, $\xi_{R_x} = \frac{P_{G_{T_2}}}{\aleph^2}$. In case of $m_2 = M_2$, the outage event can be stated as:

$$h_{R_x,2} < \frac{\tau_{R_x}(\xi_I I + 1)}{\xi_{R_x}(||\mathbb{H}||^2 \beta_{R_x})} \tag{A3}$$

The outage probability in Phase II can be expressed as follows:

$$\begin{aligned}
\mathbb{P}_2 = \mathbb{P}_r \Bigg( ||\mathbb{H}||^2 &< \left| \frac{\tau_{R_x}}{\xi_{R_x}} \right|, \\
h_{R_x} &< \frac{\tau_{R_x}(1 + \xi_I I)}{\xi_{R_x}(||\mathbb{H}||^2 \beta_{R_x} - \tau_{R_x} \sum_{m_2=j_2}^{M_2} \beta_{m_2})} \Bigg)
\end{aligned} \tag{A4}$$

where $\gamma_t = \frac{\tau_{R_x}}{\xi_{R_x}}$

$$\mathbb{P}_2 = \mathbb{P}_r \Bigg( ||\mathbb{H}||^2 < |\gamma_t|, \tag{A5}$$

$$h_{R_x} < \frac{\tau_{R_x}(1 + \xi_I I)}{\xi_{R_x}(||\mathbb{H}||^2 \beta_{R_x} - \tau_{R_x} \sum_{m_2=j_2}^{M_2} \beta_{m_2})} \Bigg)$$

$$= \int_0^{\gamma_t} f_{||\mathbb{H}||}(y)dy + \int_{\gamma_t}^{\infty} \int_0^{\infty} f_{||\mathbb{H}||^2}(y) \tag{A6}$$

$$\cdot F_{|h_{R_x}|} \left( \frac{\tau_{R_x}(1 + x\xi_I)}{\xi_{R_x}(y\beta_{R_x} - \tau_{R_x} \sum_{m_2=j_2}^{M_2} \beta_{m_2})} \right)$$

$$\cdot f_I(x)dxdy$$

where the integral in first term represents the CDF of the Rayleigh fading random variable $||\mathbb{H}||$ and is given as $1 - \exp^{\frac{-\gamma_t}{\Omega_{2,i}}}$ [31]; $f_I(.)$ denotes the probability density function (PDF) of the interference $I$. For ease of notation, let us assume that in (A6), $\eta_1 = \frac{\tau_{R_x}}{\beta_{R_x}} \sum_{m_2=j_2}^{M_2} \beta_{m_2}$, $\eta_2 = \frac{\tau_{R_x}}{\xi_{R_x}\beta_{R_x}}$, and $y - \eta_1 = z$. In addition, let us denote the integral in the second term of (A6) by $A$. Therefore, the outage probability in Phase II can now be written as follows:

$$\mathbb{P}_2 = 1 - \exp^{\frac{-\gamma_t}{\Omega_{2,i}}} + A \tag{A7}$$

In order to obtain $\mathbb{P}_2$, it is required to compute the integral in $A$. Thus, in what follows, we will solve the integral in $A$. As such, based on (A6), $A$ is given as follows:

$$A = \int_{\tilde{\gamma}_t}^{\infty} \int_0^{\infty} f_{||\mathbb{H}||}(z + \eta_1) \tag{A8}$$

$$\cdot F_{|h_{R_x}|} \left( \frac{\eta_2(1 + x\xi_I)}{z} \right) f_I(x)dzdx$$

$$= \int_{\tilde{\gamma}_t}^{\infty} \int_0^{\infty} e^{-(z+\eta_1)} \sum_{l=1}^{L} b_{m_2} e^{\frac{-\psi_l \eta_2(1+x\xi_I)}{z}} f_I(x)dzdx$$

$$= \int_0^{\infty} \sum_{l=1}^{L} b_{m_2} f_I(x) \underbrace{\int_{\tilde{\gamma}_t}^{\infty} e^{-(z+\eta_1)} e^{\frac{-\psi_l \eta_2(1+x\xi_I)}{z}} dz dx}_{A_1}$$

For ease of notation, let us denote $\left(\lambda = \psi_l \eta_2(1 + x\xi_I)\right)$ in (A8). Therefore, $A_1$ in (A8) can be expressed as follows:

$$A_1 = e^{-\eta_1} \int_{\tilde{\gamma}_t}^{\infty} e^{-z} e^{\frac{-\lambda}{z}} dz \tag{A9}$$

Using (1.211) of [32], the expression $A_1$ can be written as:

$$A_1 = e^{-\eta_1} \sum_{k_1=0}^{N_1} \int_{\tilde{\gamma}_t}^{\infty} e^{-z} \cdot \frac{(-\lambda)^{k_1}}{k_1!} \left(\frac{1}{z}\right)^{k_1} dz \tag{A10}$$

$$A_1 = \sum_{k_1=0}^{N_1} e^{-\eta_1} \frac{(-\lambda)^{k_1}}{k_1!} \int_{\tilde{\gamma}_t}^{\infty} e^{-z} \left(\frac{1}{z}\right)^{k_1} dz$$

where $N_1 = \infty$. Using (3.381.3) of [32], (A10) becomes:

$$A_1 = \sum_{k_1=0}^{N_1} e^{-\eta_1} \frac{(-\lambda)^{k_1}}{k_1!} \Gamma(1 - k_1, \tilde{\gamma}_t) \tag{A11}$$

Substituting the value of (A11) into (A8) would provide the following expression:

$$
\begin{aligned}
A &= \int_0^\infty \sum_{l=1}^{L} b_{m_2} f_I(x) \sum_{k_1=0}^{N_1} e^{-\eta_1} \frac{(-\lambda)^{k_1}}{k_1!} \Gamma(1-k_1, \tilde{\gamma}_t) dx \\
&= \sum_{l=1}^{L} \sum_{k_1=0}^{N_1} b_{m_2} e^{-\eta_1} \frac{(-1)^{k_1}}{k_1!} \Gamma(1-k_1, \tilde{\gamma}_t)(\psi_l \eta_2)^k \\
&\cdot \underbrace{\int_0^\infty (1+x\xi_I)^{k_1} f_I(x) dx}_{A_2}
\end{aligned}
\tag{A12}
$$

In order to solve $A_2$ in (A12), the PDF of interference at the receiver is required. As such, the Laplace transform of the interference PDF at the receiver is given by [18]:

$$
\mathcal{L}_I(s) = e^{-2\pi\lambda_{GT}} \sum_{p=1}^{P} \Omega_p \frac{a(1 - \mathcal{X}_1(\gamma_p) + s\gamma_p^{-a}}{1 + s\gamma_p^{-a}} \cdot \Lambda_2(c)
\tag{A13}
$$

where $\Omega_p = \omega_p e_p^{\gamma}$, $\omega_p = \frac{\Gamma(p+2)\gamma_p}{p!(p+1)^2(L_{p+1}(\gamma_p))^2}$, $L_p(.)$ is the Laguerre polynomial of degree P, and $\gamma_p$ are the roots of $L_p(.)$. $\Lambda_2(.)$ and $\mathcal{X}_1(.)$ are calculated using (A-7) and (A-10) of [2]. Based on (A13), the Laplace inverse of the interference $I$, denoted by $f_I(x)$, can be expressed as follows, from [33]:

$$
f_I(x) = \frac{e^{vx}}{T} \sum_{k_2=0}^{2N_2} {}' \Re \left[ \mathcal{L}_I(s = \mathfrak{s}) \exp\left(\frac{\iota\pi x}{T}\right) \right]
\tag{A14}
$$

where $\mathfrak{s} = v_0 + \frac{\iota\pi k_2}{T}$, $v_0 = v_1 - \frac{\log(\varsigma)}{T}$, $v, v_1 > 0$ are real numbers, $\varsigma$ is the desired relative accuracy, $T$ is a scaling parameter, $\iota = \sqrt{-1}$, $N_2$ is the number of terms used to invert Laplace transform, and the prime indicates that $k_2 = 0$ summation term is halved. Now, based on (A14), $A_2$ in (A12) can be expressed as follows:

$$
\begin{aligned}
A_2 &= \frac{1}{T} \sum_{k_2=0}^{2N_2} \int_0^\infty \Re\left[ (1+x\xi_I)^{k_1} e^{\left(v+\frac{\iota\pi}{T}\right)x} \mathcal{L}_I(\mathfrak{s}) \right] dx \\
&= \frac{1}{T} \sum_{k_2=0}^{2N_2} \Re\left[ \mathcal{L}_I(\mathfrak{s}) \underbrace{\int_0^\infty (1+x\xi_I)^{k_1} e^{\left(v+\frac{\iota\pi}{T}\right)x} dx}_{A_3} \right]
\end{aligned}
\tag{A15}
$$

It is very challenging to solve the integral in $A_3$. Therefore, we use Gauss–Laguerre quadrature to approximate $A_3$ as follows:

$$
\begin{aligned}
A_3 &= \int_0^\infty (1+x\xi_I)^{k_1} e^{\left(v+\frac{\iota\pi}{T}\right)x} dx \\
&\approx \sum_{k_3=1}^{N_3} \omega_{k_3} (1+x_{k_3}\xi_I)^{k_1} e^{\left(v+\frac{\iota\pi}{T}\right)x_{k_3}} e^{x_{k_3}}
\end{aligned}
\tag{A16}
$$

Using (A16), $A_2$ in (A15) can be written as follows:

$$
\begin{aligned}
A_2 &= \frac{1}{T} \sum_{k_2=0}^{2N_2} \sum_{k_3=1}^{N_3} \omega_{k_3} e^{x_{k_3}} (1+x_{k_3}\xi_I)^{k_1} \\
&\cdot \Re\left[ e^{\left(v+\frac{\iota\pi}{T}\right)x_{k_3}} \mathcal{L}_I(\mathfrak{s}) \right]
\end{aligned}
\tag{A17}
$$

Now, based on (A17), (A12) can be expressed as follows:

$$A = \sum_{l=1}^{L} \sum_{k_1=0}^{N_1} \sum_{k_2=0}^{2N_2} \sum_{k_3=1}^{N_3} \frac{b_{m_2}}{T} e^{-\eta_1} \frac{(-1)^{k_1}}{k_1!} (\psi_l \eta_2)^{k_1}$$
$$\cdot \, \Gamma(1 - k_1, \tilde{\gamma}_t) \omega_{k_3} e^{x_{k_3}} (1 + x_{k_3} \xi_I)^{k_1}$$
$$\cdot \, \Re\left[ \mathcal{L}_I(\mathfrak{s}) e^{(v + \frac{\iota \pi}{T}) x_{k_3}} \right]$$

(A18)

Finally, substituting $A$ from (A18) in (A7) proves the result for the outage probability in Phase II in Theorem 1. □

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
