# Peer review of "A Novel Multi-Cell Interference-Aware Cooperative QoS-Based NOMA Group D2D System"

_futureinternet, doi:10.3390/fi15040118_

Round 1

Reviewer 1 Report

This paper describes a quality of service (QoS) based cooperative NOMA-aided group D2D system (Q-CNOMA). The authors develop a mathematical model of the considered transmission scheme, evaluate its outage probability, and compare the scheme with other NOMA and OMA schemes.

I have some comments regarding this research.

1. The literature review about NOMA + D2D should be improved. It is missing such research as
[1] Zhao, Jingjing, et al. "Joint subchannel and power allocation for NOMA enhanced D2D communications." IEEE Transactions on Communications 65.11 (2017): 5081-5094.

[2] Dash, Soumya P., and Sandeep Joshi. "Performance analysis of a cooperative D2D communication network with NOMA." IET Communications 14.16 (2020): 2731-2739.

2. What is 21pt in formula 5 and in the text after it?

3. What is "k" in the second line of equation 11?

4. What kind of OMA is considered in the comparison part? What kind of power control or rate control is used there? It is an important question, because NOMA should be compared with proper rate control in OMA systems, otherwise the comparison is unfair.

Author Response

Authors’ reply to the reviewers’ comments of the manuscript futureinternet-2256582 entitled “A Novel Multi-Cell Interference Aware Cooperative QoS Based NOMA Group D2D System”

Dear Reviewer,

We have carefully revised the manuscript according to the reviewers’ comments. Accordingly, we included below our responses to the reviewers’ comments.

We thank the Editor and reviewers for their time in reviewing/managing our submission and hope that they will find this revised version and our responses satisfactory.

Reviewer 2 Report

The presented paper was devoted to well-established issues related to new spectrally effective techniques of non-orthogonal multiple access (NOMA). The authors have proposed a new Q-CNOMA method, which is based on the QoS parameters and close cooperation of terminals within the range of the base station, which can act as D2D relay. The state-of-the-art on NOMA solutions was conducted quite widely. This allowed to demonstrate that the proposed method has not yet been checked in the scientific and production environment. The results of calculations and simulations carried out by the authors of the paper show that the method is promising. The concept was presented in a legible manner, but in order to increase the value of the paper, several minor corrections and extensions should be introduced:

11.The simulation environment has not been described in principle. It must be added.

22.What specific solution was used for comparative simulation and calculations regarding OMA?

33.What effect can the mobility of devices acting as relay have on this access model? A discussion is necessary.

44.What impact on the increase in delay (quality parameter) will the number of relay nodes in the NOMA/OMA radio path have?

55.Other minor comments about need of simple corrections:

a)       row 169-170 – it should be „probability is presented in section IV.”

b)      row 178 – probably it should be “Re”. There should be given a description of the difference between Re and rx. Such a description appears only in rows 185 and 186.

c)       formula (1) – description of variable I is not present. The I as interference description only appears at the formula (10).

d)      row 194 – sentences must be combined – “After MRC, the received signal at GT2 can be written as:”.

e)      formulae (4) and (5) – indexes 25pt, 5.5pt, 21pt, 5pt are not described.

f)        formula (8) – it should be “Success in Phase I”.

Author Response

(The authors gave the same response as above.)

Reviewer 3 Report

This work formulates a model which combines NOMA and D2D transmission in a two-layer system, using relays for forwarding cooperatively the information to the cluster heads at the destination group. The outage probability and the key performance of the system are calculated. There are some issues that should be addressed to improve the quality of this paper:

1) The model is inconsistent at times. For instance, it is said that in Phase II OMA i used, but just before equation (2) the text states that NOMA is used. 

2) The mathematical notation is cumbersome and confusing. I cannot understand the subscripts 5.5pt, 25pt, 23pt, etc. appearing multiples times in the different equations. This makes the analysis unreadable, practically.

3) The numerical results are given for a very simple setting: two users, distances less than 10m, etc. This is far from any realistic network or deployment.

4) The Gaussian Poisson process is not explained in the text.

In its present form, this paper contains too many weaknesses and cannot be accepted for publication.

Author Response

(The authors gave the same response as above.)

Round 2

Reviewer 1 Report

The authors have resolved the reviewers' comments, the paper seems fine.

Reviewer 3 Report

The authors have taken care in formatting the mathematical notation and derivations in a much more clear form, and to explain the calculation and assumptions in the model properly and with the necessary motivation. I have checked the computations and seem to be correct, moreover the experimental results match well the analytical expressions.

This manuscript can be accepted for publication.